# The role of orthographic neighbourhood effects in lateralized lexical decision: a replication study and meta-analysis

Adam J. Parker[1], Ciara Egan[1,2], Jack H. Grant[1,3], Sophie Harte[1,4], Brad T. Hudson[1,5] and Zoe V.J. Woodhead[1]

[1] Department of Experimental Psychology, University of Oxford, Oxford, Oxon, United Kingdom
[2] School of Psychology, Bangor University, Bangor, Gwynedd, United Kingdom
[3] School of Psychology, University of Lincoln, Lincoln, Lincolnshire, United Kingdom
[4] Deafness, Cognition and Language Research Centre, University College London, University of London, London, United Kingdom
[5] Department of Psychology, Lancaster University, Lancaster, Lancashire, United Kingdom

## ABSTRACT

The effect of orthographic neighbourhood size (N) on lexical decision reaction time differs when words are presented in the left or right visual fields. Evidence suggests a facilitatory N effect (i.e., faster reaction times for words with larger neighbourhoods) in the left visual field. However, the N effect in the right visual field remains controversial: it may have a weaker facilitative role or it may even be inhibitory. In a pre-registered online experiment, we replicated the interaction between N and visual field and provided support for an inhibitory N effect in the right visual field. We subsequently conducted a pre-registered systematic review and meta-analysis to synthesise the available evidence and determine the direction of N effects across visual fields. Based on the evidence, it would seem the effect is inhibitory in the right visual field. Furthermore, the size of the N effect is considerably smaller in the right visual field. Both studies revealed considerable heterogeneity between participants and studies, and we consider the implications of this for future work.

## INTRODUCTION

Cognitive models are important tools for understanding how written word processing occurs in skilled readers, and how these processes may break down in individuals with developmental or acquired reading disorders. Studying how word recognition is affected by psycholinguistic variables, such as word frequency or imageability, provides a key source of evidence for developing and testing those models. The influence of orthographic neighbourhood size on lexical decision performance is one such example where the observed results have not yet been fully explained by existing models.

The orthographic neighbourhood of a word is the set of words that can be formed by substituting a single letter in a target word. For example, the neighbourhood of *sleet* is *fleet, sheet, skeet, sweet, slept, sleek*, and *sleep*. The size of this neighbourhood (i.e., the number

Corresponding author
Adam J. Parker,
adam.parker@psy.ox.ac.uk

of words in the neighbourhood; referred to as 'N' hereafter; *Coltheart et al., 1977*) has been reported to influence word identification. Several studies using the lexical decision paradigm have indicated that, for words presented in central/foveal vision, a larger N is associated with more accurate and faster responses (for reviews see *Andrews, 1997*; *Grainger, 1992*; *Mathey, 2001*; *Perea, 2015*).

This facilitative N effect may seem counterintuitive when considered within a Parallel Distributed Processing framework, such as the Interactive Activation model (*McClelland & Rumelhart, 1981*), which posits that there is lateral inhibition between competing lexical representations. This might predict that words with more neighbours would experience more lateral inhibition, resulting in slower reaction times. It has been theorised instead that the facilitation may arise due to excitatory feedback connections from the word to the letter level. When N is larger, partial activation of orthographic neighbours at the word level may thereby increase letter level activation and speed up word recognition (*Andrews, 1989*; *Andrews, 1992*; *Coltheart & Rastle, 1994*; *Whitney & Lavidor, 2005*).

The role of N during lexical access has also been examined using lateralized presentation, yielding similarly counterintuitive results. During these visual half-field experiments, words are presented in the left or right visual hemifield. Words presented to the LVF will initially be projected to the visual cortex in the right cerebral hemisphere. Words presented to the RVF will initially be projected to the visual cortex of the left cerebral hemisphere. It has been shown that in the LVF, there is a facilitative N effect so that words with more neighbours are responded to more quickly and accurately. By contrast, in the RVF the N effect is weaker or even inhibitory (*Lavidor & Ellis, 2001*; *Lavidor & Ellis, 2002*; *Perea, Acha & Fraga, 2008*; *Whitney, 2004*). This pattern of results is surprising, as the LVF shows the same direction of effect as the central visual field (i.e., facilitation); whereas the RVF, which is characterised as being more efficient for word processing due to its direct access to the language dominant hemisphere, shows an atypically weak or reversed N effect.

One specific computational model of letter-encoding that makes predictions about differential orthographic processing in each hemisphere is the SERIOL (sequential encoding regulated by inputs to oscillations within letter units) model (*Whitney, 2001*). The SERIOL model makes specific predictions about how word identification occurs from early retinotopic visual processing via abstract feature, letter and bigram levels, to a lexical word form level (see Fig. 1). At the earliest levels of the model (the retina and edge level processing in V1/V2), there is an activation gradient related to acuity, where activity is highest at central fixation and falls off into the periphery. In the left hemisphere (receiving inputs from the RVF) activity is graded from left-to-right, whereas in the right hemisphere (receiving inputs from the LVF) it is graded right-to-left. In order to preserve left-to-right serial order in the abstract letter representations, the acuity gradient in the right hemisphere has to be inverted. This gradient inversion is achieved in two stages: (1) bottom-up excitation from the edge to the feature level, which boosts activation in the right hemisphere relative to the left; and (2) left-to-right lateral inhibition at the feature level in the right hemisphere, but not the left.

*Whitney (2004)* and *Whitney & Lavidor (2005)* proposed that differences between the left hemisphere and right hemisphere processing up to the letter level may explain the

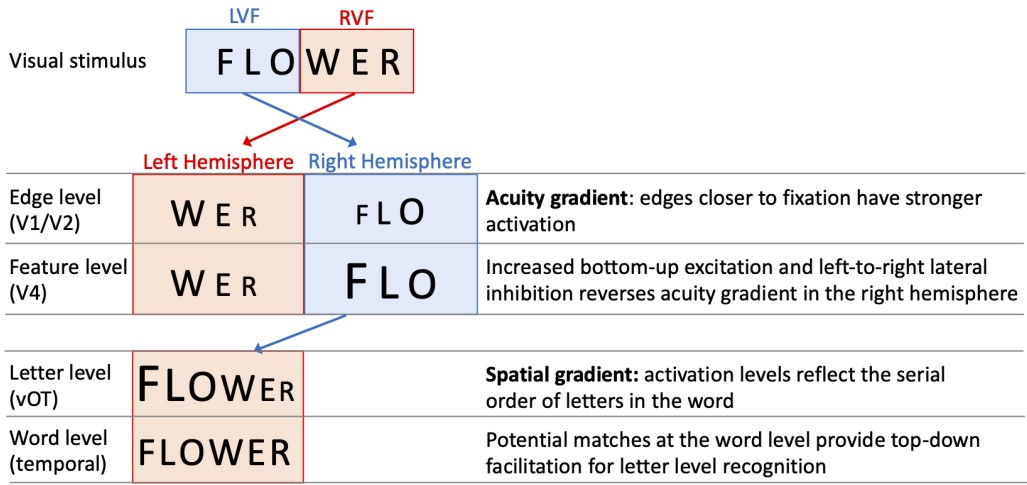

**Figure 1** **A schematic illustration of the SERIOL model, based on descriptions in *Whitney (2004)* and *Whitney & Lavidor (2005)*.** Here, fixation falls between 'O' and 'W' with letters in each visual field projecting to the contralateral hemisphere. An acuity gradient is observed at the retina and the edge level (in V1 / V2). In the right hemisphere (but not the left), there is increased bottom-up excitation and left-to-right lateral inhibition, which reverses the acuity gradient to a spatial gradient at the feature level (in V4). Feature information is transferred from the right to left hemisphere, so that at the letter level (in the ventral occipitotemporal cortex, vOT) the serial order of letters in the word is represented by the activation for each letter. Activity spreads across potential matches at the word level (in the left temporal lobe), and top-down excitation back to the letter level facilitates word recognition.

differential effects of N across hemispheres. Under visual half field viewing conditions, words presented to the LVF extend further than normal into parafoveal vision, and the usual processes of excitation and lateral inhibition may be insufficient to fully invert the acuity gradient. As a result, the spatial gradient of letters for LVF words may be imprecise, and require greater than usual top-down support from the word level to resolve word recognition. As a result, there will be a greater than usual facilitatory N effect, as words with more neighbours will produce stronger top-down support from the word to letter level. Put more simply, the SERIOL model suggests that there is relatively poor orthographic encoding in the LVF, which is offset by increased top-down orthographic facilitation from the word level back to the letter level. More top-down orthographic facilitation means a stronger N effect, and hence we observe a facilitative N effect in the right hemisphere (*Whitney, 2004*).

In contrast, the SERIOL model posits that for words presented in the RVF, letter processing is more efficient as the acuity gradient matches the serial order of letters in the word. This results in more efficient orthographic encoding in the left hemisphere. This means that top-down excitatory feedback has a weaker effect in the left hemisphere (*Whitney & Lavidor, 2005*), making N effects smaller or negligible.

Given SERIOL's predictions for the role of N in each hemisphere, we would expect empirical studies which examine effects of N and visual field on accuracy and reaction time to yield a significant interaction. This interaction would point towards a facilitative N effect in the LVF but not in the RVF. Indeed, several studies have reported facilitative effects of

[1]Accuracy in these studies is typically high and variability is low. While some studies report effects of visual field on accuracy (with greater accuracy in the RVF relative to the LVF; *Perea, Acha & Fraga, 2008*), others do not (*Lavidor & Ellis, 2001*). There is also little evidence to conclude that N, or the interaction between N and visual field, yield strong effects on accuracy.

[2]This effect size was estimated from Fig. 1 in *Lavidor, Johnston & Snowling (2006)* using the metagear package (version 0.6; *Lajeunesse, 2016*).

N in the LVF (*Ellis, 2004*; *Lavidor & Ellis, 2001*; *Lavidor & Ellis, 2002*; *Lavidor, Johnston & Snowling, 2006*; *Mano et al., 2010*; *Perea, Acha & Fraga, 2008*; *Whitney & Lavidor, 2005*).[1] Out of these studies, some report weaker facilitatory effects in the RVF (*Lavidor & Ellis, 2001*; *Lavidor, Johnston & Snowling, 2006*; *Whitney & Lavidor, 2005*), while others report inhibitory effects (*Lavidor & Ellis, 2002*; *Mano et al., 2010*; *Perea, Acha & Fraga, 2008*). The largest inhibitory N effect in the RVF to date was reported in a sample of Spanish speaking participants. *Perea, Acha & Fraga (2008)* reported a very small (−2 ms) facilitative effect in the central visual field (CVF), a much larger facilitative effect in the LVF (−23 ms), and a strong inhibitory effect in the RVF (28 ms). This contrasts the equally sizeable -24 ms facilitatory effect[2] in the RVF reported by *Lavidor, Johnston & Snowling (2006)* for typical English readers. The inhibitory effect observed by Perea et al. in the RVF may, to some extent, reflect differences across writing systems. Compared to English, Spanish is a more shallow, transparent orthography. *Whitney & Lavidor (2005)* noted that word identification in shallow orthographies may depend more on phonology-to-orthography than word-to-orthography connections. As a consequence, activation from high-N words in Spanish may be less than in English; a prediction that was confirmed by *Carreiras, Perea & Grainger (1997)*. Furthermore, substitution neighbours in English tend to occur disproportionately in initial letter positions whereas they are more distributed in Spanish. Despite these differences in orthographic depth, it is clear that studies do report inhibitory N effects in the RVF for English participants (*Mano et al., 2010*; *Lavidor & Ellis, 2002*, Experiment 2).

The extent to which the N effect is weakly facilitatory or inhibitory in the RVF (i.e., left hemisphere) is of interest then to models that make predictions based on each hemisphere's involvement in word recognition. To further examine the role of N under lateralized presentation, and to add weight to the claim that N effects are inhibitory in the RVF, the current work describes two studies of N effects during lateralized lexical decision. The first was an online replication of *Perea, Acha & Fraga (2008)*, who reported facilitative effects of N in the LVF and inhibitory effects in the RVF. In anticipation of the results, we replicate Perea et al.'s findings and report that N effects differ across visual fields in a sample of English speaking participants. We subsequently conducted a systematic review and meta-analysis of the published literature to determine both the direction and size of N effects in the RVF.

## STUDY 1: AN ONLINE REPLICATION STUDY

We first set out to examine whether we could replicate the pattern of results reported by *Perea, Acha & Fraga (2008)* in an online replication study with English speaking participants. The experiment was pre-registered on the Open Science Framework prior to the commencement of data collection. We pre-registered three criteria for a successful replication: (1) reduced errors and faster reaction times in the CVF relative to the LVF and RVF for word targets; (2) reduced errors and faster reaction times in the RVF relative to the LVF for word targets; (3) a facilitative N effect in the LVF and an inhibitory effect in the RVF in reaction times for word targets. The registration form, materials, data, and R scripts can be found at https://osf.io/vknbp.

## Method

### Participants

*A priori* power analysis using pilot data and the SimR package (version 1.0.5; *Green & MacLeod, 2016*) in R (*R Development Core Team, 2016*) suggested that 30 participants would provide sufficient power (power >80%) to detect the critical N by visual field interaction in reaction time. The critical effect size used in these simulations was 15 ms, which is more conservative than the effect size reported by *Perea, Acha & Fraga (2008)*. To reach this sample size, 45 native English speakers were recruited through Prolific Academic (https://www.prolific.co/); a web-based platform for recruiting and screening paid research participants according to experimenter-specified criteria. Participants were right-handed, had normal or corrected-to-normal vision, no significant hearing loss, history of neurological disease, head injury, epilepsy, or diagnosis of language and/or reading impairment. Thirteen participants were removed from the study as their score on the Edinburgh Handedness Inventory was below 80 (*Oldfield, 1971*). This cut-off was chosen to match *Perea, Acha & Fraga (2008)*. Two further participants were removed from the study as they scored below chance when completing the lexical decision task in at least one of the three visual field conditions. The remaining 30 participants (18 female) had a mean age of 29.0 years ($SD_{age} = 8.12$ years).

This study was approved by the University of Oxford's Medical Sciences Inter-divisional Research Ethics Committee (reference number: R67385/RRE001) and conducted in accordance with the principles of the Declaration of Helsinki. Participants gave informed written consent and received compensation at a rate of £10/hour for their participation.

### Materials

Stimuli were selected using the same restrictions as *Perea, Acha & Fraga (2008)*. We selected 120 word and 120 non-word targets. All targets had two syllables and were 5 letters in length. For word targets, 60 had few substitution neighbours ($M_N = 0.7$, range$_N$: 0–1) and 60 had many substitution neighbours ($M_N = 9.5$, range$_N$: 7–12). N was acquired using the English Lexicon Project (*Balota et al., 2007*). Word targets did not differ in zipf frequency scores (low N: $M_{zipf} = 3.1$, $SD_{zipf} = 0.82$; high N: $M_{zipf} = 3.1$, $SD_{zipf} = 0.77$; Mann Whitney $U = 1768.00$, $p = .869$), obtained from the SUBTLEX-UK database (*van Heuven et al., 2014*). One hundred and twenty non-word targets were selected from the English Lexicon Project (*Balota et al., 2007*) and had either no substitution neighbours or one substitution neighbour.

### Design and procedure

The experiment consisted of two tasks: (1) the Edinburgh Handedness Inventory, (2) the visual half-field task. The visual half-field task was within-subjects and utilised a 2 (lexicality: word, non-word) ×2 (N: low, high) ×3 (visual field: RVF, LVF, CVF) design.

The study was administered via Gorilla (https://gorilla.sc/; *Anwyl-Irvine et al., 2020*). Participants were instructed to sit approximately 50 cm from the screen. Each trial began with a fixation cross presented centrally for 400 ms. Participants were instructed to keep their eyes on the central fixation point. Then, a target was presented for 150 ms to the left, right, or at the position of the fixation point. Previous attempts at using Gorilla to run
similar visual half-field tasks involving rapid presentation of stimuli yielded good test-retest reliability (*Parker et al., 2020*).

Stimuli were presented in black lowercase Tahoma font on a white background. Gorilla's scaling tool was used to ensure that stimulus size and position were kept consistent despite differences in participants' screen sizes. Font was scaled to 250% equating to 28pt font, and stimuli were presented at a displacement of 2.5° of visual angle when presented to the left or right of fixation. Participants were instructed to press one of two buttons on the keyboard to indicate whether the string was a legitimate English word or not ("S" for word or "K" for non-word). They were instructed to do so as quickly and accurately as possible. The next trial started contingent on the participant's response. Stimuli were divided into three counterbalanced lists with each participant responding to 120 words and 120 non-words in a random order. The whole session took approximately 12 minutes.

### Data analysis

Error rates were analysed using a generalized linear mixed-effects model (GLMM) fitted to binomial data using the *glmer()* function and logit link from the lme4 package (version 1.1-21; *Bates et al., 2019*). Reaction times were analysed with linear mixed-effects models (LMM) using the *lmer()* function from the lme4 package. Both models adopted an identical structure: $dv \sim VF + N + VF:N + (1 \mid participant) + (1 \mid word)$. The *contr.sum* function was used to implement summed-to-zero contrasts for N, constituting a main N effect. The *contr.helmert()* function was used to implement Helmert contrasts for visual-field. This enabled us to contrast means for the LVF and RVF directly before contrasting the combined mean of LVF and RVF with the mean for the CVF. This series of contrasts resulted in the intercept corresponding to the grand mean. For each model, we report regression coefficients (*b*), 95% confidence intervals (*95% CI*), standard errors (*SE*), and *t/z* values. We consider as statistically significant those cases where $|t/z| > 1.96$.

While we considered all trials for error rate analysis, only correct responses were considered for reaction time analysis. Trials were excluded if reaction times were less than 250 ms or greater than 1,200 ms.[3] Following the removal of these cases, we identified and removed outliers using *Hoaglin & Iglewicz*'s (*1987*) procedure. For each participant (across all conditions) we calculated the difference between first and third quartiles of the response time data. Outliers were then defined as those where reaction times were 1.65 times this difference above the third and below the first quartiles. This enabled us to exclude trials with particularly long responses which may have resulted from delayed loading times. These procedures led to the removal of 12.3% of trials.

### Results

### Error rates

Error rates are shown in Table 1. The GLMM fitted to error rates indicated that there were fewer errors in the CVF relative to the mean of the RVF and LVF (see Table 2). There were fewer errors in the RVF than the LVF. The remaining fixed effects did not significantly impact error rates.

---

[3]Removal procedures were optimised using pilot data prior to pre-registration. After applying these removal procedures and log transformation, RTs approximated a gaussian distribution.

**Table 1  Mean error rates and reaction times for word targets as a function of neighbourhood size (N) and visual field.**

|  | Low N | | | High N | | |
|---|---|---|---|---|---|---|
|  | **LVF** | **CVF** | **RVF** | **LVF** | **CVF** | **RVF** |
| Error rates (%) | 43.2 (49.57) | 16.0 (36.69) | 26.7 (44.26) | 47.3 (49.97) | 19.7 (39.78) | 30.0 (45.86) |
| Reaction times (ms) | 774.0 (158.10) | 684.5 (165.98) | 715.6 (178.09) | 749.2 (162.94) | 676.0 (157.63) | 726.1 (185.81) |

**Notes.**
Standard deviations are shown in parentheses.

**Table 2  (G)LMM results for error rates and reaction time analysis.**

| Measure | Fixed effect | *b* | 95% CI | *SE* | *z/t* |
|---|---|---|---|---|---|
| Error rates | (intercept) | **−1.124** | **[−1.436, −0.812]** | **0.159** | **−7.06** |
|  | VF (LVF –RVF) | **−0.471** | **[−0.568, −0.373]** | **0.050** | **−9.46** |
|  | VF (CVF –LVF and RVF) | **−0.418** | **[−0.483, −0.352]** | **0.033** | **−12.53** |
|  | N | −0.140 | [−0.344, 0.064] | 0.104 | −1.34 |
|  | VF (LVF –RVF) × N | −0.037 | [−0.133, 0.060] | 0.049 | −0.74 |
|  | VF (CVF –LVF and RVF) × N | −0.033 | [−0.098, 0.031] | 0.033 | −1.00 |
| log-RT | (intercept) | **2.859** | **[2.836, 2.882]** | **0.011** | **250.23** |
|  | VF (LVF –RVF) | **−0.014** | **[−0.018, −0.009]** | **0.002** | **−5.93** |
|  | VF (CVF –LVF and RVF) | **−0.015** | **[−0.017, −0.012]** | **0.001** | **−12.04** |
|  | N | 0.002 | [−0.004, 0.007] | 0.003 | 0.59 |
|  | VF (LVF –RVF) × N | **−0.005** | **[−0.009, −0.001]** | **0.002** | **−2.00** |
|  | VF (CVF –LVF and RVF) × N | −0.001 | [−0.003, 0.007] | 0.001 | −0.55 |

**Notes.**
Significant model terms are shown in bold.

### Reaction times

The LMM fitted to log-transformed reaction times indicated that reaction times were faster in the CVF relative to the mean of the RVF and LVF (meeting our first criterion for successful replication). Furthermore, reaction times were faster in the RVF than the LVF (our second criterion for replication). While the main N effect was not significant, the interaction between N and visual field (LVF –RVF) indicated that the N effect differed in the RVF and LVF. In the LVF, high N targets were responded to 24.7 ms faster than low N targets. In contrast, in the RVF high N targets were responded to 10.5 ms slower than low N targets (see Figs. 2A–2C). This pattern of effects was near identical to that reported by *Perea, Acha & Fraga* (*2008*; see Figs. 2D–2F). The second N by visual field interaction (CVF –LVF and RVF) did not significantly impact reaction time. This indicates that when N effects were summated across the LVF and RVF, the N effect did not statistically differ from the N effect in the CVF.

### Exploratory analyses

In addition to our pre-registered analyses, we conducted exploratory analyses looking at the N effect at the participant level and how lateralisation on the visual half-field task correlated with the magnitude of the N effect.

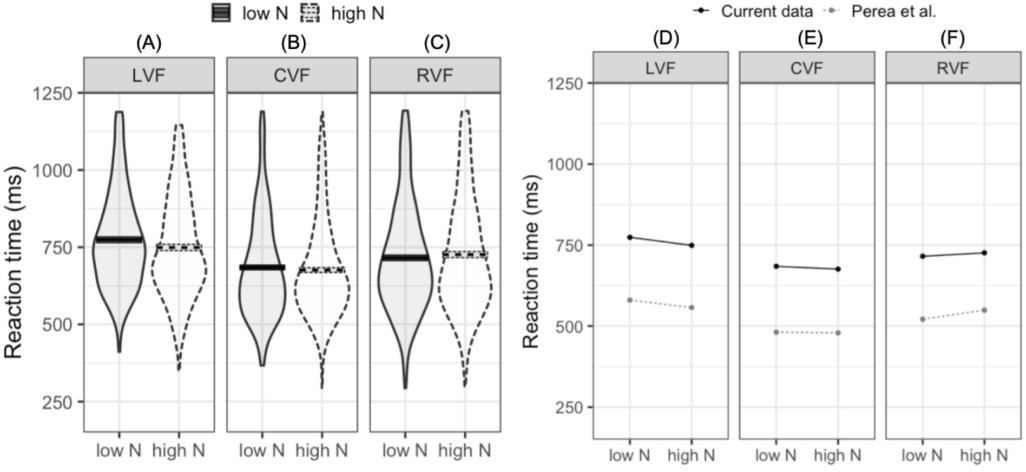

**Figure 2 Plots for pre-registered analyses.** Left: violin plots of RT for high- and low-N words in the left visual field (A), central visual field (B), and right visual field (C). The horizontal band reflects the mean in each experimental condition. Solid black lines correspond to low-N and dashed grey lines correspond to high-N. Right: Means across experimental conditions in the current study and *Perea, Acha & Fraga (2008)*. Both datasets show a facilitatory N effect in the left visual field (D), a small facilitatory effect in the central visual field (E), and an inhibitory N effect in the right visual field (F). LVF, left visual field; CVF, central visual field; RVF, right visual field.

To date, no study investigating the N effect during lateralized lexical decision has examined the extent to which individual participants show facilitatory or inhibitory effects of N in each visual field. Out of the 30 participants in the current study, 20 showed a facilitatory effect in the LVF, 14 showed a facilitatory effect in the CVF, and 11 showed a facilitatory effect in the RVF. The reaction time differences for individual participants are shown in Fig. 3A, with the numerical difference for each participant reported in the (Table S1). Furthermore, only 12 out of the 30 participants showed both a facilitatory N effect in the LVF and an inhibitory effect in the RVF, indicating that while this pattern of results is consistent at the population level, there are likely to be subtle individual differences in N effects across hemispheres.

One potential explanation for the observed differences in the magnitude of the N effect across each visual field is that the N effect is influenced by the extent to which participants are lateralized for language. It seems plausible that those who are left hemisphere language dominant may show one pattern of N effects while those who show bilateral, or even right hemisphere dominance, may show weaker effects of N or even a reversed pattern. To explore this potential explanation, we calculated laterality indices (LIs) using correct responses to both high and low N words in either the LVF or RVF: we assume that participants who are left-hemisphere dominant for language should show a RVF advantage (and therefore would respond faster to RVF words), whereas participants who are right hemisphere dominant would show a LVF advantage.

LIs were calculated using the formula: $100 \times (\text{Right} - \text{Left})/(\text{Right} + \text{Left})$, where Right refers to the mean reaction time in the RVF and Left refers to the mean reaction time in

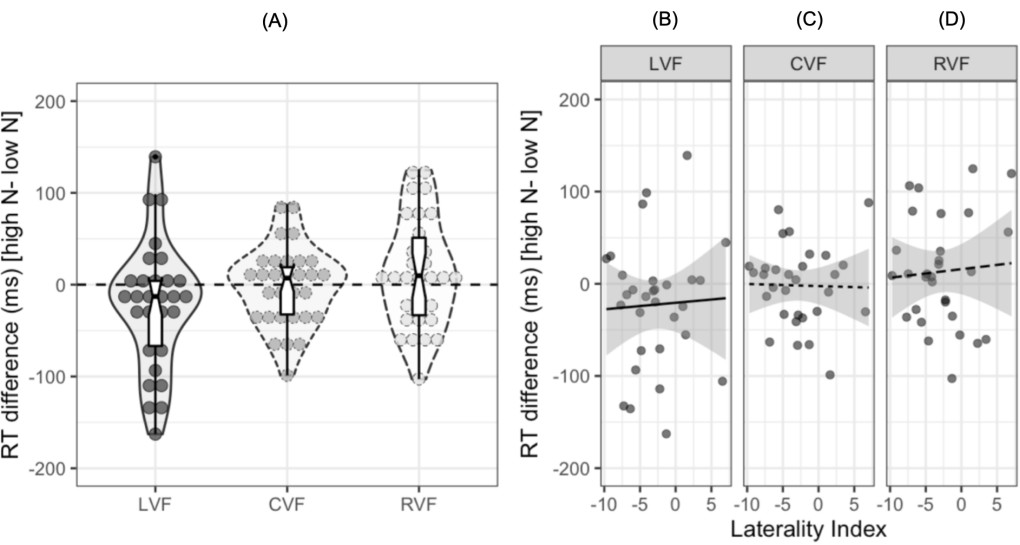

**Figure 3  Plots for exploratory analyses.** (A) Reaction time differences (ms) [high N - low N] in each visual field. Dots represent individual participants' data while the boxplot shows the median and interquartile ranges. The horizontal dashed black line represents zero, or no N effect. (B), (C), and (D) show scatterplots for the participants' reaction time difference (ms) and laterality index in the left, central, and right visual fields respectively. The solid and dashed lines represent the relationship between the two variables. The shaded grey region represents 95% CIs.

[4]We repeated these analyses when including participants with scores below 80 on the Edinburgh Handedness Inventory to increase variability in LIs. Again, a series of Pearson's product-moment correlations indicated that the N effect was not significantly correlated with LIs in the LVF, $r = -.14$, 95% CI [$-.42$, .17], $p = .382$, CVF, $r = -.04$, 95% CI [$-.27$–.33], $p = .819$, or RVF, $r = .07$, 95% CI= [$-.37$–.24], $p = .657$.

the LVF. A negative LI on this scale indicates lateralisation to the left hemisphere and a positive LI indicates lateralisation to the right hemisphere. The mean LI was $-2.8$ ($SD = 4.27$) and a one-sample $t$-test indicated that there was significant lateralisation to the left hemisphere at the population level, $t (29) = -3.26$, 95% CI [$-4.42$, $-1.23$], $p = 0.001$. A series of Pearson's product-moment correlations indicated that the N effect was not correlated with LIs in the LVF, $r = .04$, 95% CI [$-.32$, .40], $p = .818$, CVF, $r = -.02$, 95% CI [$-.38$–.34], $p = .908$, or RVF, $r = .07$, 95% CI [$-.30$, .42], $p = .727$.[4] Figure 3D shows scatterplots of these results.

## Study 1: Discussion

In Study 1 we conducted a replication of *Perea, Acha & Fraga (2008)* with a sample of English speaking participants. Analysis of error rates and reaction times met our three criteria for successful replication. We reported: (1) reduced errors and faster reaction times in the central visual field relative to the left and right visual fields for word targets; (2) reduced errors and faster reaction times in the right visual field relative to the left visual field for word targets; (3) a facilitative N effect in the left visual field and an inhibitory effect in the right visual field in reaction times for word targets. This pattern of results replicates several prior studies (*Lavidor & Ellis, 2002*; *Mano et al., 2010*; *Perea, Acha & Fraga, 2008*) and confirms predictions from the SERIOL model regarding the interaction between N and visual fields.

The observation of a visual field by N interaction at a population level can be interpreted within the context of the SERIOL model. Facilitatory effects of N in the LVF (i.e., right
hemisphere) occur as top-down orthographic activation offsets poorer encoding. This implies weak bottom-up input from the letter to word level and increased top-down input to letter nodes due to global activation in the lexicon. By contrast, more efficient letter encoding in the RVF (i.e., left hemisphere) results in a reduced need for top-down input from the word to letter levels.

However, this explanation from the SERIOL model fails to capture the individual differences that we observed in the current study when we considered N effects in each visual field. Our data suggest that only 12/30 participants show strictly facilitatory effects of N in the LVF and inhibitory effects in the RVF, indicating that participant-level characteristics must influence the pattern of this effect. To start to examine which participant-level characteristics influence the strength of the N effect, we explored the extent to which lateralisation on our visual half-field task influenced the size of the N effect across visual fields. The data suggest that there is no statistically reliable relationship. However, it is important to note here that the study was not designed for the assessment of individual differences as participants completed different versions (counterbalanced lists) of the task meaning that the obtained correlations likely reflect some between item variation. Thus, this is one potential avenue for future research.

It is important to note that the −8 ms facilitatory N effect in the CVF we observed is considerably smaller than N effects typically reported under central presentation. For instance, *Pollatsek, Perea, & Binder (1999)* reported a −26 ms facilitative N effect under central presentation. Several explanations for these observed differences are possible. First, the current study presented stimuli for 150 ms. By contrast, in lexical decision experiments involving only central presentation, stimuli are typically visible until a response is made. As such, the N effect may be reduced when stimulus presentation is shorter. Such an explanation would explain why *Perea, Acha & Fraga (2008)* reported a small −2 ms N effect in the CVF. However, such a possibility would be difficult to reconcile with the large −33 ms facilitative CVF N effect reported by *Lavidor & Ellis (2002)* which adopted the same 150 ms stimulus presentation duration. A second explanation for this reduced N effect in the CVF relates to stimuli selection. In a review of the published literature, *Andrews (1997)* concluded that facilitatory effects of N are restricted to low-frequency items. Thus selecting a relatively high proportion of high-frequency items has the potential to diminish N effects under central presentation. Indeed, when applying a median split to zipf frequency, the N effect was more pronounced for low-frequency words (−13.7 ms) than for high-frequency words (−1.0 ms).

In comparison to the published literature on lateralized N effects, the reaction times in the current study were considerably longer; as illustrated in Fig. 2B. The same is true when comparing our results to other lateralized lexical decision experiments (e.g., *Lavidor & Ellis, 2001*). Such differences in reaction times between lab- and web-based studies have been observed across a range of experimental paradigms (e.g., *Hilbig, 2016; Semmelmann & Weigelt, 2017*). Several explanations for this observation exist and relate to environmental factors such as hardware and/or software. First, participants engaging in online studies may be using slower computers than those taking part in lab-based studies and this might influence the absolute millisecond-based measurements negatively. Second, such
reaction time differences may be caused by inaccuracies in the browser engine, as they are not optimised for stimulus presentation and the recording of response times. Regarding population level effects, it is reassuring to know that *Hilbig (2016)* reported near identical frequency effects in a lexical decision task that was carried out under lab- and web-based conditions. This suggests that although reaction times collected online may be longer, they can still accurately capture well-established effects at the millisecond level.

In sum, our replication study adds weight to the claim that effects of N differ across visual fields. However, one question that remains is whether the N effect is reliably different from zero in the RVF. As the current replication study was powered to detect the interaction between N and visual fields, we decided to conduct a systematic review and meta-analysis to address this quandary.

# STUDY 2: A SYSTEMATIC REVIEW AND META-ANALYSIS

Despite the consensus that N effects differ across visual fields, the direction and magnitude of the N effect appear far more consistent in the LVF than the RVF. While most studies report a facilitatory effect in the LVF, the nature of the effect seems somewhat controversial in the RVF. Therefore, we conducted a systematic review and meta-analysis to synthesise the available data and provide the first statistical analysis examining whether the N effect was reliably different from zero in the LVF and RVF.

## Method

The protocol for the systematic review and meta-analysis was pre-registered using the Non-Interventional, Reproducible, and Open (NIRO) Systematic Reviews guidelines (*Toper et al., 2020*). No deviations were made from the registered protocol. The registration form, data, and R scripts can be found at https://osf.io/7ekyw.

### *Literature search*

The search of the literature followed the PRISMA guidelines (*Moher et al., 2009*) and covered all articles published between 1st January 1997 and 1st June 2020. Searches were conducted on the 18th June 2020 using several databases containing published and unpublished literature. The following bibliographic databases of published literature were searched: ProQuest, ScienceDirect, and Web of Science. The grey/unpublished literature were searched using: Ethos, OpenGrey.eu, Open Science Framework Pre-Prints, PsychArXiv, and DissOnline.de. Specifically, we searched the following terms in the title, topic, and abstract fields of each database: ("orthographic neighbo*" AND "hemisphere"), ("orthographic neighbo*" AND "hemifield"), ("orthographic neighbo*" AND "laterali*"), ("orthographic neighbo*" AND "visual field"), ("orthographic neighbo*" AND "visual-field"), ("orthographic neighb*" AND "Visual half field"), and ("orthographic neighb*" AND "visual half-field"). The reference list of key papers were also checked (*Andrews, 1997*; *Ellis, 2004*; *Lavidor & Ellis, 2001*; *Lavidor & Ellis, 2002*; *Perea, Acha & Fraga, 2008*; *Whitney, 2004*).

The study selection process is illustrated in Fig. 4. The abstracts of articles identified through database searches were screened for duplicates. All duplicates were then removed.

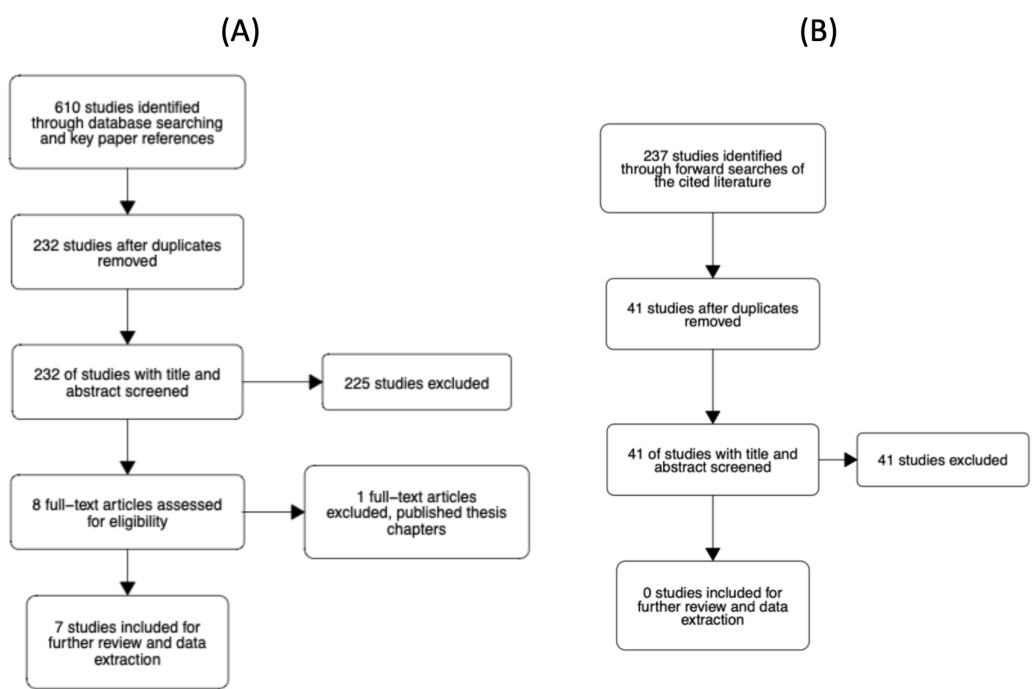

**Figure 4** **Flowcharts of the literature search process.** (A) Our initial search of the literature. (B) Our forward search of the cited literature from eligible studies.

Six reviewers then screened titles and abstracts of 232 unique articles using the metagear package (version 0.6; *Lajeunesse, 2016*), which was also used to store and manage bibliographic data throughout the review process. Reviewers assessed whether the article looked potentially eligible for inclusion and responded with yes, no, or maybe. Each article was screened twice by a different reviewer. The agreement rate was 94.4%. In the case of no agreement, a review was conducted by a third reviewer. From the 232 unique articles, seven were considered eligible for further screening. Four articles were rated as 'maybe' relevant by reviewers, which prompted screening of the methods sections. Of these four, one article was considered eligible for inclusion. Thus, there were eight eligible articles in total.

Each of the eight articles was then evaluated against the inclusion criteria presented in the Supplemental Materials. In short, the studies had to examine the N effect during a lateralized lexical decision and include reaction time as a dependent variable. At this stage, one article was removed as it was a doctoral thesis where the relevant chapters had been published and the corresponding article had already been deemed eligible for inclusion. In addition to reviewing the full text, we checked the Retraction Watch Database (http://retractiondatabase.org/RetractionSearch.aspx?) to ensure these articles had not been retracted. None of the articles that passed a full-text review were indicated to have been retracted.

For the 7 included papers we conducted a forward search of the cited literature. In total, 237 studies were identified. After removing duplicates and articles that had previously been

screened, 41 additional unique articles were identified. Each article was screened twice by a set of four reviewers. The agreement rate was 93%. For 3 articles where there was disagreement, a third screen was conducted. None of the 41 articles were rated as eligible for inclusion in the meta-analysis.

To summarise, our initial search and screening strategies led to seven articles being identified for use in the current meta-analysis (*Fiset & Arguin, 1999*; *Lavidor & Ellis, 2001*; *Lavidor & Ellis, 2002*; *Lavidor, Johnston & Snowling, 2006*; *Mano et al., 2010*; *Perea, Acha & Fraga, 2008*; *Whitney & Lavidor, 2005*).

### Data extraction and effect size calculation

Of these seven articles, two reported multiple experiments (*Lavidor & Ellis, 2002*; *Whitney & Lavidor, 2005*). One of these (*Lavidor & Ellis, 2002*) included a between items manipulation of word length. Rather than aggregate across this manipulation, we treated each level of the manipulation as a separate experiment. Thus, data extraction was conducted for 11 experiments by two reviewers. In addition to extracting values such as sample size, language of stimuli, etc., reviewers extracted mean reaction times for low- and high-N words in both the LVF and RVF. This could be completed for all but one experiment (*Fiset & Arguin, 1999*). We contacted the authors of the experiment in an attempt to gain summary statistics and/or raw data. However, the authors could not locate the data, thus the experiment was removed from the meta-analysis. For the remaining experiments, both reviewers extracted identical mean reaction times across conditions and experiments.

As each experiment adopted a within-subjects design, it was necessary to obtain an estimate of the correlation between reaction times for high- and low-N words in each visual field. As none of the experiments included in the meta-analysis reported correlations between reaction times for low- and high-N words across visual fields, we contacted the first, last, and corresponding authors for each of the six articles requesting both the full and aggregate data. The authors indicated that they no longer had access to the data or we received no reply within one month. Thus, we could not obtain an estimate of the correlation between mean reaction times in the high- and low-N condition in either visual field from the published literature. As such, we decided to use our replication study to derive estimates of these correlations. From this dataset, the correlation between reaction times in the low- and high-N condition was $r = .90$ in the LVF and $r = .79$ in the RVF. It is important to note that these correlations are stronger in magnitude than those typically observed in psychological science. This will of course have implications for the outcome of our meta-analysis; as the correlation coefficient increases, the estimate of the variance for each study decreases and precision increases. As such, we felt it necessary to repeat the meta-analysis assuming a number of values for $r$ and report these in the Supplemental Materials.

To estimate the size of the N effect across visual fields, reaction times in the low N condition were subtracted from scores in the high N condition. The variance of the effect size was calculated using Formulas 12.8–12.9 from *Borenstein et al. (2009)*. The variance of

each study ($V$) was calculated as:

$$V = \frac{S_{dif}^2}{n}$$

where $S_{diff}$ is the standard deviation of the mean difference for that study and $n$ is its sample size. $S_{diff}$ was calculated as:

$$S_{dif} = \sqrt{S_1^2 + S_2^2 - 2 \times r \times S_1 \times S_2}$$

where $S_1$ is the standard deviation of the high N condition, $S_2$ is the standard deviation of the low N condition, and $r$ is the correlation between the means in the high and low N conditions. A summary of the effect sizes for the included experiments are shown in Table 3.

### Publication bias

We assessed publication bias by presenting effect sizes against the inverse of their standard error in a funnel plot (*Sterne et al., 2011*). Symmetrical funnel plots indicate an absence of bias while asymmetrical funnels plots indicate the presence of bias. For the LVF, the funnel plot indicated no clear evidence of asymmetry/publication bias (see Fig. S1). To statistically assess asymmetry we conducted Egger's regression (*Egger et al., 1997*), which indicated no significant evidence for funnel plot asymmetry in the LVF ($t$ (9) $= -.23$, $p = .821$). Inspection of the funnel plots for the RVF indicated potential asymmetry (see Fig. S2). However, the Egger's regression for the RVF indicated no significant evidence for funnel plot asymmetry ($t$ (9) $= -2.06$, $p = .070$).

### Quality assessment

The quality of each study was assessed by two reviewers via the Appraisal Tool for Cross-Sectional Studies (AXIS; *Downes et al., 2016*). The AXIS tool contains 20 questions grouped under introduction, method, results, discussion, and other. As some of the questions appeared more relevant to clinical work, we used only 15 items here. Ratings for each of the six published articles appear in the S2. Scores ranged from 6–11 for Reviewer 1 and from 7–11 for Reviewer 2.

It is noteworthy that each study used an incredibly similar methodology. All studies relied on right-handers drawn from a university population. Furthermore, the experimental setup was highly consistent across studies. For instance, the presentation time and displacement of stimuli did not vary considerably between studies. Most studies presented stimuli for 150 ms at 2.5° eccentricity.

The studies differed in how they operationalise high- and low-N. *Lavidor & Ellis (2002)* reported that words had a mean N of 9.5 and 1.0 in their high- and low-N condition. *Perea, Acha & Fraga (2008)* reported that the mean N of words in the high- and low-N conditions were 8.4 and 0.3 respectively. It may, therefore, be of interest to examine how the between-study differences in N thresholds influence the magnitude of the effect size across visual fields.

**Table 3** Experiments included in the meta-analysis.

| Study | Language | Sample size | Word/cond. | Visual field | |
| --- | --- | --- | --- | --- | --- |
| | | | | LVF | RVF |
| *Lavidor & Ellis (2001)* | English | 24 | 20 | −49.0 | −1.0 |
| *Lavidor & Ellis (2002)*, Exp 1 | English | 24 | 20 | −35.0 | 1.0 |
| *Lavidor & Ellis (2002)*, Exp 2, 4-letters | English | 24 | 16 | −29.0 | 19.0 |
| *Lavidor & Ellis (2002)*, Exp 2, 5-letters | English | 24 | 16 | −13.0 | 14.0 |
| *Lavidor & Ellis (2002)*, Exp 2, 6-letters | English | 24 | 16 | −44.0 | 20.0 |
| *Whitney & Lavidor (2005)*, Exp 1 | English | 19 | 13 | −25.0 | −3.0 |
| *Whitney & Lavidor (2005)*, Exp 2 | English | 25 | 13 | −22.0 | 3.0 |
| *Lavidor, Johnston & Snowling (2006)* | English | 11 | 20 | −36.1 | −23.9 |
| *Perea, Acha & Fraga (2008)* | Spanish | 30 | 20 | −23.0 | 28.0 |
| *Mano et al. (2010)* | English | 26 | 20 | −46.0 | 15.0 |
| Current data | English | 30 | 20 | −24.7 | 10.5 |

**Notes.**

Positive effects indicate inhibitory effects for N while negative effects indicate facilitatory effects. Lavidor & Ellis, Exp 2 contained targets of different word length. Rather than aggregate over each word length, we included each as a separate study.
[1] Study 1 this article.

### Data analysis

*Meta-analysis.* Data were synthesised using random-effects meta-analyses. The analyses were conducted in R using the *rma()* function from the metafor package (version 2.4-0; *Viechtbauer, 2010*) and compared the N effect in each visual field separately. This enabled us to test the size of the N effect directly in each visual field, rather than compare studies for the presence of an interaction as a meta-regression would permit.

For each model, we used the Sidik-Jonkman estimator (*Sidik & Jonkman, 2002*) to estimate the between-study variance and applied the Hartung-Knapp adjustment (*Hartung & Knapp, 2001*) when estimating effect sizes and confidence intervals. To statistically quantify the magnitude of heterogeneity, we examined the $I^2$ and Cochrane's Q. Because the correlation between reaction times for high- and low-N words in each visual field were estimated based on one study, we additionally conducted sensitivity analyses with alternative estimates of this correlation, where $r = .2$, $.5$, and $.8$ constituting to a small, medium, and large effect and report these in the Supplemental Materials. In sum, the outcome of the meta-analysis was consistent irrespective of the size of the correlation coefficient used to calculate the variance of each study.

*Meta-regression.* We conducted exploratory meta-regression using the *rma()* function from the metafor package to examine how the strength of the N manipulation influenced N effects across visual fields.

### Results
#### Meta-analysis

The results from the meta-analysis are shown in Table 4 and Fig. 5. The estimated N effect in the LVF was −31.8 ms, indicating that the N effect is facilitatory in the right hemisphere. In contrast, the estimated effect in the RVF was 10.8 ms, indicating that N is

**Table 4 Effect size estimate of the N effect in the LVF and RVF and the 95% confidence intervals from the meta-analysis model.** The $I^2$ and $Q$ values statistically quantify the heterogeneity of the studies included.

| Visual field | k | Mean ES (ms) | 95% CI | p-value | $I^2$ (%) | Q |
|---|---|---|---|---|---|---|
| LVF | 11 | −31.8 | [−39.82, −23.69] | <.001 | 59.43 | $Q(10) = 38.11, p < .001$ |
| RVF | 11 | 10.8 | [2.84, 18.84] | .013 | 81.75 | $Q(10) = 69.35, p < .001$ |

**Notes.**

LVF, left visual field; RVF, right visual field; k, number of studies included in the analysis; Q, Cochrane's Q; $I^2$, percentage of variance that can be attributed to between-study heterogeneity.

inhibitory in the left hemisphere. Not only was the direction of the effect different across visual fields, so was the size of the effect. Scrutiny of the $I^2$ statistic indicated moderate heterogeneity between studies comparing N effects in the LVF (*Higgins et al., 2003*). The $I^2$ statistic was larger for studies comparing the N effect in the RVF, indicating increased variation in outcomes for the RVF relative to the LVF. This is an issue we return to in our meta-regression.

### Meta-regression

A series of exploratory meta-regressions were conducted to examine how the strength of the N manipulation influenced the N effect in each visual field. The strength of the N size manipulation was calculated by subtracting the mean low N rating for stimuli from the high N rating. For example, *Lavidor & Ellis (2002)* report that words had a mean N of 9.5 and 1.0 in their high- and low-N condition. Thus, the strength of the N manipulation was defined as 9.5–1.0 = 8.5. We fit a meta-regression model to the LVF data which indicated that the strength of the N manipulation did not influence the N effect in the LVF, $b = .78$, 95% CI [−3.28–4.85], $SE = 2.07$, $p = .706$. An identical model was fit to the RVF data, which yielded a similar pattern of results where the strength of the N manipulation did not influence the N effect, $b = −3.34$, 95% CI [−7.13–.46], $SE = 1.94$, $p = .085$.

## Study 2: Discussion

This meta-analysis aimed to synthesise the available evidence on orthographic neighbourhood size (N) during lateralized lexical decision to estimate whether the N effect is facilitatory or inhibitory in the LVF and RVF. Based on the data, we can conclude that the N effect is facilitatory for words presented in the LVF (i.e., right hemisphere) and inhibitory for words presented in the RVF (i.e., left hemisphere). This indicates that orthographic information is processed differently across the two cerebral hemispheres. We will return to a discussion of the implications of these results in the General Discussion.

What is surprising from this meta-analysis is the level of heterogeneity, despite all studies adhering to a very similar methodology. Inspection of the Cochrane's Q and $I^2$ indicated substantial to large heterogeneity around the estimates of N effects in each visual field. When conducting the systematic review, it became clear that the strength of the N manipulation fluctuated between studies. However, the size of the N manipulation between studies did not significantly influence the N effect across visual fields.

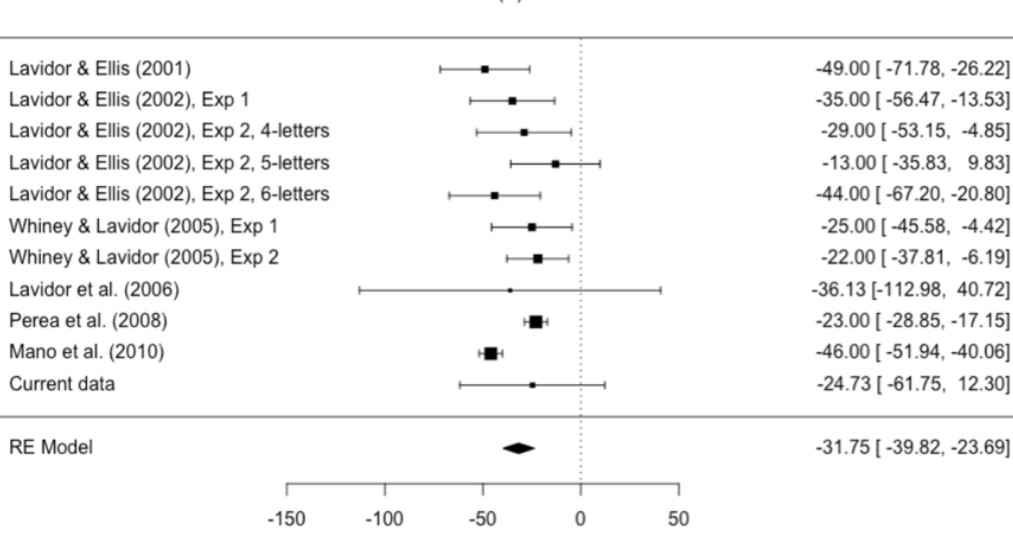

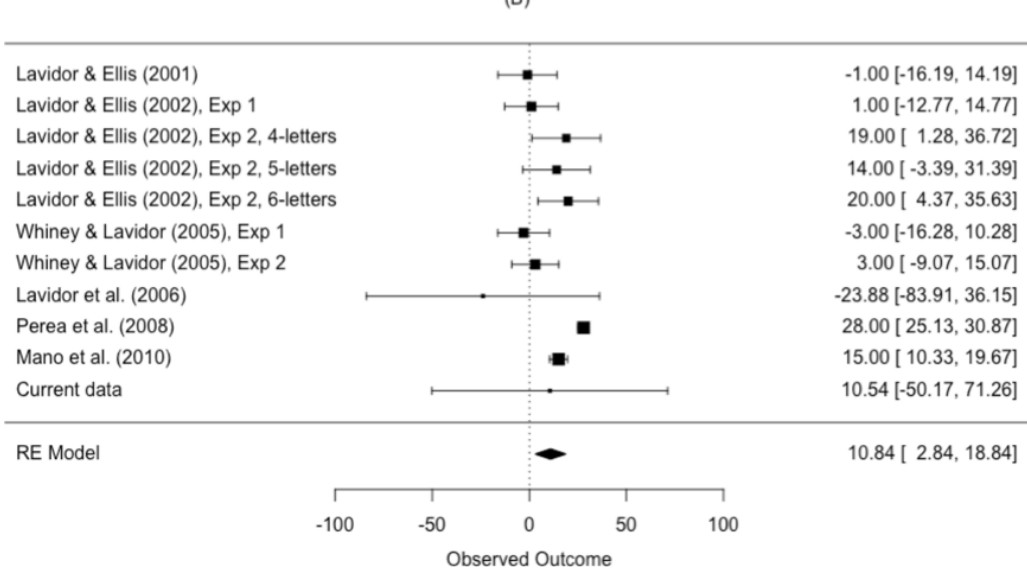

**Figure 5  Forest plot of the N effect in the (A) left visual field and the (B) right visual field.** Plotted are the observed study means and 95% CI. The size of squares is proportional to the weight of each study in the analysis (i.e., the inverse of the within-study variance of the sampling distribution).

The source of between study heterogeneity may instead result from differences in stimulus selection. For instance, the size of the N effect in each study will be influenced by factors such as the presence of stimuli with body neighbours and/or a word's frequency of occurrence in natural language. Body neighbours are orthographic neighbours that differ in relation to the first letter position (e.g., *h_ouse* / *m_ouse*) (*Andrews, 1997*). Body neighbours
typically rhyme and have a facilitative effect on lexical decision times (*Ziegler & Perry, 1998*). Thus, a study that has many words with body neighbours may encounter different N effects than a study with fewer words with body neighbours. Similarly, as N effects appear stronger for low-frequency items (*Andrews, 1997*) it may be that studies which use a higher proportion of low-frequency items N effects of a different magnitude than a study using few low-frequency items. Other sources of variation may relate to environmental factors such as contrast and luminosity that are not maintained across experiments. Below, in the General Discussion, we speculate on several participant-level explanations for this.

## GENERAL DISCUSSION

Over the last two decades, there have been a number of studies on orthographic neighbourhood effects under lateralized lexical decision, but with inconclusive results. Despite the apparent evidence of a facilitatory effect of having many neighbours in the LVF, the direction and magnitude of N effects in the RVF has been somewhat controversial. While almost all studies report an interaction between N and visual field, some studies report the N effect to be facilitatory in the RVF (e.g., *Lavidor, Johnston & Snowling, 2006*) while others report it to be inhibitory (e.g., *Mano et al., 2010*; *Perea, Acha & Fraga, 2008*). The novel contributions of our current work can be summarised in two general points. First, in a replication study, we reported evidence of an interaction between N and visual field confirming that the main N effect differs between the left and right visual fields. Second, in a systematic review and meta-analysis, we reported the N effect to be strongly facilitative in the LVF and inhibitory in the RVF. We discuss these findings within the context of models of word recognition and consider future directions for this line of research.

These findings are compatible with the SERIOL model (*Whitney, 2004*). The SERIOL model proposes that letter encoding for words presented in the LVF is inefficient or imprecise, because there is a mismatch between the acuity gradient (based on the distance from fixation) and the serial order of letters in the word (see Fig. 1). As a result, top-down support from the word level is required to disambiguate the word. Spreading activation among potential word level targets results in more top-down support for words with larger neighbours: hence, N size has a facilitative effect for words presented in the LVF. By comparison, letter encoding is efficient in the RVF, as the letters' serial order matches their proximity to fixation. Because letter identification is rapid in the RVF, there is less chance for top-down processing to influence letter encoding; explaining negligible effects of N in the RVF. Instead, lateral inhibition between neighbours at the word level may result in competition, and slower reaction times for high N words. The results reported here support the view that the N effect is inhibitory in the RVF at the population level, with Study 2 showing for the first time that this effect is reliably different from zero.

The Multiple Read-Out Model (MROM; *Grainger & Jacobs, 1996*) also offers an explanation for these findings. Under this account words in a lexical decision task are identified when the critical activation level is reached on one of the appropriate response criteria, namely M or $\Sigma$. The M criterion refers to activation of the specific whole-word

orthographic representation, whereas Σ refers to overall activation in the orthographic lexicon. According to the MROM the N effect is dependent on which criteria is used to generate most responses, with M leading to an inhibitory effect, and Σ leading to a facilitative effect. Due to the differential processing efficiencies for orthographic input in the two hemispheres, it may be that different criteria are preferentially used across the visual fields (cf. *Perea, Acha & Fraga, 2008*). More specifically, the M criteria may be used more frequently for words presented in the RVF due to more efficient processing in the left hemisphere. In contrast, the less efficient processing in the right hemisphere (LVF) may lead to greater use of the Σ criterion. This would then lead to N having an inhibitory effect for words presented in the RVF and a facilitative effect for words presented in the LVF, which is consistent with the results of the meta-analysis.

It is worth highlighting that although our findings indicate a facilitative N effect in the LVF and an inhibitory effect in the RVF at the *population* level, this pattern was rarely observed within individual participants. In fact, there was considerable heterogeneity in the pattern of N effects seen at the *participant* level. This suggests that there are additional factors that influence the N by visual field interaction, which vary between individuals. This may reflect the fact that the two effects (facilitation in the LVF, and inhibition in the RVF) result from two different mechanisms (inefficiency of letter encoding, and lateral inhibition at the word level). It is plausible that individual differences may affect these two mechanisms independently.

One important avenue for future research is the extent to which the precision and redundancy of participants' orthographic representations influence word processing under lateralized presentation. Research into lexical quality (as indexed by reading and spelling ability) in other areas of psycholinguistics has broadened our understanding of how participant level characteristics (typically reading and spelling ability) influence response times in lexical decision (*Andrews, 2012*; *Andrews & Lo, 2012*; *Andrews & Hersch, 2010*) and eye movements during reading (*Drieghe et al., 2019*; *Parker & Slattery, 2020*; *Veldre & Andrews, 2014*). While no study has explicitly examined the influence of lexical quality on lateralized N effects, *Andrews & Hersch (2010)* investigated how masked neighbour priming effects are influenced by written language proficiency. They reported that poor spellers showed facilitation from a higher frequency orthographic neighbour prime (e.g., *jury*, *FURY*), whilst good spellers showed inhibition. These results indicate that high-quality lexical representations (as indexed by spelling ability) afford rapid activation of the briefly presented primes, which inhibit orthographically similar representations. However, it remains unclear if the precise orthographic knowledge indexed by spelling ability will influence the N effect in a divided visual half-field paradigm. It seems plausible that those with the most robust lexical representations may show inhibitory effects regardless of visual field while those with weaker lexical representations may show inhibition and facilitation, thus explaining some of the between-participant differences in the pattern of N effect in the LVF and RVF. An investigation of the N by visual field interaction in skilled and less-skilled readers would not only inform our interpretation of the between-participant differences in N effects, but would inform computational models of word recognition

that have classically ignored individual differences in the processing of lexical information across cerebral hemispheres.

Alternatively, it may be the case that individual differences in the N size effect in left and right visual fields relates to strength or direction of language lateralization. While approximately 90% of right-handers are left hemisphere dominant for language (*Knecht et al., 2000a*), a substantial number of individuals demonstrate bilateral or right language dominance (*Knecht et al., 2000b*). Investigating the relationship between language lateralization and N effects may be informative in understanding the locus of the N effect: if the N effect relates solely to the match between visual acuity and serial letter order (as predicted by the SERIOL model), it should not vary according to language lateralization. However, if it relates instead to the delay caused by callosal transfer from one hemisphere to another (the mechanism used to explain the RVF advantage), lateralization should have an influence on the strength or direction of the N effect.

Finally, we would like to note that as Study 1 was conducted online, we could not control for various environmental factors such as luminance, contrast, the participants' attention or distance from the screen, and these may explain the variability in our data. It is reassuring nonetheless that we managed to replicate the previous literature despite potential room for increased noise. One way to add credibility to our novel findings would be to examine the variability between participants with data collected under lab condition.

## CONCLUSIONS

Two decades ago it was first reported that words with many orthographic neighbours were processed differently in each visual field (*Lavidor & Ellis, 2001*). Since then, numerous studies have investigated this interaction with many claiming that the N effect is faciliatory in the LVF. However, the N effect in the RVF remains somewhat controversial. The size of the N effect in the RVF has differed considerably, with estimates ranging from $-23.9$ to $28.0$ ms. Until now no study has sought to provide a synthesis of the data to determine whether the effect is reliably different from zero. Evidence from our meta-analysis clearly indicates that this effect does exist in the RVF; however, it is less sizable than the N effect in the LVF. These results will have important implications for models of word recognition that specify different contributions of each cerebral hemisphere in word processing. Additionally, we provide the first look at individual differences in the N effect and report that only a subset of participants show both a facilitative N effect in the LVF and an inhibitory N effect in the RVF. This opens an avenue for future work to explore the role of individual differences in the processing of N under lateralized presentation.

At a more general level, the studies reported within this work provide evidence that online behavioural research methods provide a viable tool to examine lateralized word identification. The pattern of results across both studies provide evidence that word recognition processes differ across the cerebral hemispheres. As such, it will be important for future studies of word recognition and sentence reading to make specific predictions regarding hemispheric differences in word identification if we intend on fully understanding the cognitive processes underlying the processes of written language.

## ACKNOWLEDGEMENTS

Portions of this work were presented at the 2020 online meeting of the Experimental Psychology Society and the 61st Annual Meeting of the Psychonomic Society. We would like to thank Dorothy Bishop for continued input and guidance with this body of work. We would also like to thank Manuel Perea and an anonymous reviewer for their insightful comments on a prior manuscript.

### Funding

This work was supported by an Advanced Grant awarded by the European Research Council (Project 694189—Cerebral Asymmetry: New directions in Correlates and Etiology –CANDICE). The funders had no role in study design, data collection and analysis, decision to publish, or preparation of the manuscript.

### Grant Disclosures

The following grant information was disclosed by the authors:
Advanced Grant awarded by the European Research Council: 694189.

### Competing Interests

The authors declare there are no competing interests.

### Author Contributions

- Adam J. Parker conceived and designed the experiments, performed the experiments, analyzed the data, prepared figures and/or tables, authored or reviewed drafts of the paper, reviewed articles for meta-analysis, and approved the final draft.
- Ciara Egan analyzed the data, prepared figures and/or tables, authored or reviewed drafts of the paper, reviewed articles for meta-analysis, and approved the final draft.
- Jack H. Grant and Brad T. Hudson analyzed the data, authored or reviewed drafts of the paper, reviewed articles for meta-analysis, and approved the final draft.
- Sophie Harte analyzed the data, prepared figures and/or tables, authored or reviewed drafts of the paper, reviewed articles for meta-analysis, and approved the final draft.
- Zoe V.J. Woodhead conceived and designed the experiments, analyzed the data, prepared figures and/or tables, authored or reviewed drafts of the paper, reviewed articles for meta-analysis, and approved the final draft.

### Human Ethics

The following information was supplied relating to ethical approvals (i.e., approving body and any reference numbers):

University of Oxford's Medical Sciences Inter-divisional Research Ethics Committee approved this research (reference number: R67385/RRE001).

## Data Availability

Data and code are available at OSF: Parker, Adam J, Zoe V.J. Woodhead, and Dorothy V.M. Bishop. 2021. "Lexical Competition and Hemispheric Specialisation." OSF. January 21. https://osf.io/vknbp.

## Supplemental Information

Supplemental information for this article can be found online at http://dx.doi.org/10.7717/peerj.11266#supplemental-information.

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
