# Peer review of "The role of orthographic neighbourhood effects in lateralized lexical decision: a replication study and meta-analysis"

_PeerJ, doi:10.7717/peerj.11266_

## Round 0.1 · original submission · Minor Revisions

I have received reviews from Manuel Perea and one other prolific psycholinguistics researcher. Their comments appear below. Both are highly positive about the manuscript, as am I. It is well-written and researched and represents the highest standards in open science. Bravo! Both reviewers have made suggestions for improving the manuscript. I believe their suggestions should not pose a considerable barrier to the subsequent publication of this manuscript. I will highlight a couple points here.

Dr. Perea requests a slight change to the approach in the introduction, in hopes of making the manuscript more appealing to folks beyond the “word nerd” community. I support this suggestion, as a reframing could bring about renewed interest in this research topic, but more importantly, expose a broader audience to this master class in open science practices.

Reviewer 2 has requested that you also share analyses without the strict exclusions related to handedness. While your choices were justified, given the prior literature, there is value in exploring the limits of this phenomenon, especially given the new approach to virtual data collection. This could be a nice analysis to include in the supplementary materials alongside your other non-registered analyses. I will not insist on this addition, but the data could be informative for future virtual psycholinguistics (and laterality) research.

Thank you for submitting your work to PeerJ. It was a pleasure to read, and I look forward to the revision. On round two, I am optimistic that I will be able to make a decision without seeking guidance from our reviewers, who have been generous with their time.

·

Basic reporting

This is a very interesting manuscript on hemispheric differences in the size/direction of neighborhood effects during word identification. It contains two interesting studies, the first study (empirical) was conducted to solve some previous discrepancies in the literature, and the second is a systematic review that offers both the whole idea and also some guidelines for future research. The manuscript was also very well written and elegantly analyzed.

The literature background is excellent. However, when indicating the reviews of neighborhood effects, the range of the cited reviews was from 1992 to 2001 (i.e., 20 years ago!). This offers the reader that the field is not (highly) active any more. Perhaps citing a more recent review of neighborhood reviews would be a good idea. There is a relatively recent review in 2015 on “Neighborhood effects in visual-word recognition and reading” published as a review chapter in The Oxford Handbook on Reading edited the Pollatsek and Treiman (doi:10.1093/oxfordhb/9780199324576.013.7)

Experimental design

The experiment (Experiment 1) has been competently conducted. The potential limitations (e.g., due to the online testing) have been cited and discussed. The systematic review has been competently conducted.

Validity of the findings

The findings have important implications for reading researchers.

Additional comments

I believe the the authors need to add an Introductory paragraph in the Introduction that engages the readers of the paper to eagerly continue reading it. It goes straight to what an orthographic neighbor is. This may work for the "word nerd" community, but not outside. The authors should be able to handle this issue easily.

Also, at the end of the manuscript, the authors need to offer the broader implications of their findings.

Reviewer 2 ·

Basic reporting

This is a bold trial to run visual field studies via online platform. The writing style is clear and professional, and the authors managed well the tricky work of merging empirical and review writing.

The introduction and discussion should pay some attention to the fact that the original Perea et al study from 2008 was conducted in Spanish, a language with much shallower orthography than English therefore the N effects might differ due to different information loaded at the beginning and ends of words.

Experimental design

It is encouraging to see that the visual field by N effects was replicated, even though it was not possible to control the actual distance, refresh rates, keyboards differences etc when subjects conducted the study at their homes. Most subjects were excluded due to lower handedness score rather than technical issues resulting from home-based sessions. Since the replication differs on crucial aspects from the original (different language), I wonder whether applying a strict handedness criteria is really justified? I think it is important to analyze all subjects to further validate the possibility of remote lab testing without direct control of actual visual fields, different hardware etc.

Validity of the findings

The replication supports the validity of remote lab tests.

The meta-analysis is good, the small number of studies is not surprising and represents well the research in this field.

Additional comments

Parker et al. did a great job and it is clear that on top of performing good science, they are interested and invest a lot in meta-science and the tools that enable transparent and honest science. I enjoyed very much seeing how our old ideas were tested and receiving new and brilliant interpretation. I wish that Carol Whitney could have seen this! She would love it. Alas Carol sadly died in 2019 after a brief illness.

---

## Round 0.2 · accepted · Accept

I have read your revised manuscript, and I am satisfied that you have addressed the reviewers' critiques. I am happy to accept your manuscript for publication at PeerJ. Bravo!

As we move this into the production phase, I do want to note one typo that should be fixed. On page 6, line 14, "phonology-to-orgthaphy" should be "phonology-to-orthography."

As an aside, once upon a time, I carried out a series of experiments exploring laterality effects for a variety of psycholinguistic variables (including orthographic neighborhood), examining how those effects may change when reading handwritten words. It was a mess. Nobody can read handwritten words that fall any distance from the fovea. I cannot recommend using handwritten words in a visual half-field experiment unless you are willing to employ a titration procedure on target durations.